# Morpho-Agronomic Characterisation of Runner Bean (*Phaseolus coccineus* L.) from South-Eastern Europe

**Lovro Sinkovič [1],\*** , **Barbara Pipan [1]** , **Mirjana Vasić [2]** , **Marina Antić [3]** , **Vida Todorović [3]** ,
**Sonja Ivanovska [4]** , **Creola Brezeanu [5]** , **Jelka Šuštar-Vozlič [1]** and **Vladimir Meglič [1]**

1   Crop Science Department, Agricultural Institute of Slovenia, 1000 Ljubljana, Slovenia;
    barbara.pipan@kis.si (B.P.); jelka.sustar-vozlic@kis.si (J.Š.V.); vladimir.meglic@kis.si (V.M.)
2   Institute of Field and Vegetable Crops, 21000 Novi Sad, Serbia; mirjana.vasic@ifvcns.ns.ac.rs
3   Faculty of Agriculture, University of Banja Luka, 78000 Banja Luka, Bosnia and Herzegovina;
    marina.antic@igr.unibl.org (M.A.); vida.todorovic@agro.unibl.org (V.T.)
4   Department of Genetics and Plant Breeding, Faculty of Agricultural Sciences and Food, 'Ss.
    Cyril and Methodius' University in Skopje, 1000 Skopje, North Macedonia; s_ivanovska@yahoo.com
5   Vegetable Research and Developments Station Bacau, 600401 Bacău, Romania; creola.brezeanu@yahoo.com
\*   Correspondence: lovro.sinkovic@kis.si; Tel.: +386-1-280-5278

**Abstract:** In South-Eastern Europe, the majority of runner-bean (*Phaseolus coccineus* L.) production is based on local populations grown mainly in home gardens. The local runner-bean plants are well adapted to their specific growing conditions and microclimate agro-environments, and show great morpho-agronomic diversity. Here, 142 runner-bean accessions from the five South-Eastern European countries of Slovenia, Bosnia and Herzegovina, Serbia, North Macedonia and Romania were sown and cultivated in their respective countries and characterised using 28 quantitative and qualitative morpho-agronomic descriptors for *Phaseolus* spp. based on inflorescences, leaves, plants, pods and seeds. For each of these morpho-agronomic descriptors, the accessions can be classified into two or three specific groups. The highest correlations were observed within the fluorescence, seed and pod traits. The highest variability, at 76.39%, was between the different countries, representing different geographic origins, while the variability within the countries was 23.61%. Cluster analysis based on these collected morpho-agronomic data also classified the accessions into three groups according to genetic origins. The data obtained serve as useful genetic information for plant breeders for the breeding of new bean varieties for further studies of the morpho-agronomic traits of the runner bean.

**Keywords:** runner-bean collection; accessions; descriptors; south-eastern Europe; morpho-agronomic traits

## 1. Introduction

The runner bean (*Phaseolus coccineus* L.) is an important perennial *Phaseolus* species that originated in the tropical humid uplands of Mesoamerica [1]. It is one of the five *Phaseolus* spp. that were domesticated in Mesoamerica, and it is the third-most economically important worldwide, after *Phaseolus vulgaris* L. and *Phaseolus lunatus* L. [2]. Outside the region of origin, the runner bean is often cultivated as an annual crop for the dry seeds or is harvested as immature green pods for human consumption [3]. Cultivated runner bean was domesticated independently within two centres of diversity, which has giving rise to two gene pools: Mesoamerican and Andean [4]. In the 16th century, the runner bean was introduced into Europe, where it then spread from Spain to Italy, and then to other European countries [5,6]. Genetic diversity, population structure and phylogenetic studies have indicated that the European and Mesoamerican gene pools can be clearly differentiated, which confirms

the hypothesis that Europe can be considered as a secondary diversification centre for *P. coccineus* [6,7]. Taxonomists have described three botanical varieties of the runner bean: the white-flowered type known as *P. coccineus* var. *albiflorus*; the red-flowered type known as *P. coccineus* var. *coccineus*; and the type with white and red flowers known as *P. coccineus* var. *bicolor* [8–10]. Furthermore, the colour of the flowers is correlated to the colour of the stems and seeds, and to the seed colour pattern [9]. The runner bean is broadly distributed in South-Eastern Europe with different vernacular names of the species, which include 'Scarlet bean', 'Multiflora bean', 'Turkish bean' and 'Keber', among others [11,12]. The economic importance of the runner bean is, however, lower than that of the common bean, *P. vulgaris*.

Over recent decades, several thousand runner-bean accessions have been collected in different parts of Europe and stored in national gene banks, although the characterisation data that are needed for future use and breeding are still lacking and/or are not easily accessible. The information about individual accessions, and particularly those conserved in ex-situ collections, is often poor, reducing the frequency and efficiency of its use, and likewise, the ultimate benefits this information can provide [13]. The collections of runner beans from south-eastern European countries of, for example, Slovenia, Bosnia and Herzegovina, Serbia, North Macedonia and Romania, consist of seeds of populations and landraces that were collected in these territories, and they represent an invaluable germplasm source. The European Gene-Bank Integrated System (AEGIS) provides for the accessions in European collections to be conserved in accordance with agreed quality standards, and to be made readily available and easily accessible for breeding and research. According to the annual programme of the individual plant gene banks, only a small number of runner-bean accessions are regenerated each year, and comprehensive characterisation and evaluation of the large-scale collections have not been performed to date. This lack of characterisation of the *P. coccineus* germplasm restricts its use as a donor species for interspecies hybridisation, and consequently limits its use in other *Phaseolus* spp. breeding programmes, such as those for the common bean [1].

In these South-Eastern European countries, the majority of runner-bean production is based on local landraces, populations and varieties that are grown on a small scale, mainly in home gardens [12,14,15]. These populations are well adapted to their specific growing conditions and microclimate agro-environments, and they show great genetic and morphological diversity. The role of beans in the human diet arises not only due to their high protein content, but also to the functional properties of some of their components that have been shown to contribute to reduced risk of several diseases [16–19]. The unripe runner-bean seeds in the immature pods and the dry seeds are mostly used for the preparation of traditional dishes, such as salads or stews [15]. Boiled runner beans have a distinctive chestnut flavour, which can be acceptable particularly for traditional dishes within each country. As runner-bean plants have a predominately indeterminate or climbing type of growth, they require climbing support during their cultivation. Climbing forms of the runner bean are often grown in home gardens as ornamental vine plants, because of their pleasing inflorescences of white and/or red flowers [20].

The objective of the present study was to evaluate runner-bean accessions from gene banks of these five South-Eastern European countries, sown and cultivated in their respective countries to represent their different geographic origins. The evaluation included 28 morpho-agronomic traits related to their inflorescences, leaves, plants, pods and seeds. This information based on *Phaseolus* spp. descriptors continues to be the first step in the description, evaluation and classification of large runner-bean collections conserved in South-Eastern Europe.

## 2. Materials and Methods

### 2.1. Plant Material

The collection used in the present study comprised five distinct groups of runner-bean accessions (*P. coccineus* L.) of the individual countries obtained from their national plant gene banks: from Slovenia

(44 accessions), Bosnia and Herzegovina (10 accessions), Serbia (nine accessions), North Macedonia (64 accessions) and Romania (12 accessions). Additionally, three commercial varieties of runner bean were included in the evaluation for each country: 'Emergo' (Semenarna Ljubljana, Slovenia), 'Bonela' (Austrosaat, Austria) and 'Darko' (Federal Institute of Agriculture in Sarajevo, Bosnia and Herzegovina). Altogether, the runner bean collection used in the present study comprised 139 accessions plus three commercial varieties, i.e., 142 accessions. All of these accessions are listed in Supplementary Table S1.

The individual runner-bean accessions from each national plant gene bank, together with the three commercial varieties, were sown and cultivated in experimental fields in their respective country, i.e., 44 accessions from Slovenia plus three commercial varieties were cultivated in Slovenia, 10 accessions from Bosnia and Herzegovina plus three commercial varieties in Bosnia and Herzegovina; the same was done for the remaining accessions and countries. Field trials were performed during the 2017 growing season, according to the established production techniques of their country of origin. In Slovenia, this was at the Agricultural Institute of Slovenia, in Jablje (304 m a.s.l.; 46.151°N 14.562°E), in Bosnia and Herzegovina, at the Faculty of Agriculture, University in Banja Luka (163 m a.s.l.; 44.775°N 17.214°E), in Serbia, at the Institute of Field and Vegetable Crops in Novi Sad (80 m a.s.l.; 45.251°N 19.851°E), in North Macedonia, at 'Ss. Cyril and Methodius' University of Skopje in Pehcevo (1002 m a.s.l.; 41.762°N 22.887°E), and in Romania, at the Vegetable Research and Developments Station in Bacau (163 m a.s.l.; 46.586°N 26.954°E). The climate conditions for all five of the experimental sides during the growing season in 2017 are shown in Supplementary Figure S2.

The trials were carried out under open-field conditions with three beanpoles per accession, and four holes with two seeds per beanpole, thus as 24 seeds per accession, with 0.80 m between individual beanpoles. The seeds were disinfected before sowing according to the standard protocol, in two phases: (i) thermotherapy, where the seeds were incubated in a heating chamber with hot air at 43 °C for 72 h; (ii) chemical disinfection, which was performed on-field prior to sowing, with the seeds submerged in 5% sodium hypochlorite for 3 min, or for 1 min for the white seeds due to their greater susceptibility to chemical disinfection. During their growth, at the relevant plant developmental stages, the individual morpho-agronomic traits for each runner-bean accession were evaluated descriptively.

## 2.2. Morpho-Agronomic Characterisation

The runner-bean collection was evaluated using a total of 28 quantitative and qualitative morpho-agronomic descriptors for *Phaseolus* spp. that were related to their inflorescence, leaf, plant, pod and seed data. Several *Phaseolus* spp. descriptors were applied, as harmonised by the European Cooperative Programme for Plant Genetic Resources (ECPGR) and designed by the Community Plant Variety Office [21,22], the International Board for Plant Genetic Resources [23], and the PHASELIEU project [24]. All of the qualitative descriptors were assessed visually, while the quantitative descriptors were measured using digital Vernier calliper readings (to 0.1 mm), tape measure readings (to 1 cm) or electronic laboratory scale weights (to 0.01 g).

Data on the morpho-agronomic traits listed below were collected according to the descriptor list for *Phaseolus* of the ECPGR [25], and the full descriptions are summarised in Supplementary Table S2. The inflorescence descriptors were for days from sowing to 50% flowering [DAYSBLOS], colour of flower standard [BLOSVEX] and colour of flower wings [BLOSWING]. The leaf descriptors were for leaf shape [LEAFSHAPE], leaf colour for anthocyanins [LEAFANTHO] and leaflet length [LEAFLENGTH]. The plant descriptors were for plant growth habit [TYPE] and plant height [HEIGHT]. The seed-pod descriptors were for pod colour of the fully expanded immature pods [PODCOLIMM], pod suture strings [STRING], pod colour at physiological maturity [PODCOLMAT], pod length at physiological maturity [PODLENGTH], pod width at physiological maturity [PODWIDTH], pod cross-section at physiological maturity [PODCROSS], pod curvature at physiological maturity [PODCURV], days to 90% maturity [MATURITY], number of locules per pod [PODLOCULES] and number of seeds per pod [PODSEED]. The seed descriptors were for seed coat colour [SEEDCOLOUR], seed coat pattern [SEEDPATCHAR], brilliance of seed [BRILLIANCE], seed length [SEEDLENGTH], seed

height [SEEDHEIGHT], seed width [SEEDWIDTH], longitudinal seed shape [SEEDSHAPE], colour of seed hilum [SEEDHILUM], colour of seed hilum ring [SEEDHILUMRING] and 100-seed weight [SEEDHKM].

Additionally, for the three traits of BLOSVEX, BLOSWING and SEEDCOLOUR, two colours occurred within the same runner-bean accessions, and therefore, these were assigned as BLOSVEX1, BLOSVEX2, BLOSWING1, BLOSWING2, SEEDCOLOUR1 and SEEDCOLOUR2.

### 2.3. Statistical Analysis

The numbers of accessions with specific morpho-agronomic traits are shown as frequency distribution histograms for the whole runner-bean collection evaluated, not including the commercial varieties. For the statistical analysis, a total of 153 observations were included, as 139 runner-bean accessions and three commercial varieties per participating country (data for the 'Darko' variety in Bosnian and Herzegovinian are not available). The differences between the runner-bean accessions in the collection were analysed using a general linear model procedure and least-squares mean tests (Statgraphics Centurion XVI 2009), with a 0.05 level of significance. The statistics included mean, minimum, maximum, standard deviation (SD), coefficient of variation (CV) and analysis of variance (ANOVA). Correlations were obtained as Pearson's correlation coefficients. Principal component analysis and cluster analysis were performed to define the most influential morpho-agronomic traits that discriminated among the accessions. Biplots for the runner-bean collection were constructed for three principal components to illustrate the influence of these morpho-agronomic traits. Dendrograms were constructed to combine the individual variables into larger clusters using Ward's method and squared Euclidian distances. A box and whisker plot for 28 morpho-agronomic traits of this runner-bean collection from South-Eastern Europe is shown as Supplementary Figure S1.

## 3. Results

### 3.1. General Variability of the Runner-Beans from South-Eastern Europe

A total of 28 quantitative and qualitative morpho-agronomic descriptors related to inflorescences, leaves, plants, pods and seeds were evaluated for these 142 runner-bean accessions that originated from South-Eastern European countries. Table 1 summarises the statistics of the 12 quantitative morpho-agronomic traits for this runner-bean collection. Summary data for morpho-agronomic traits according to the countries of origin of the three commercial varieties included in the study are presented in Supplementary Table S3. The data on the individual country evaluations are shown in Supplementary Table S4. For the means, for inflorescences, the time to 50% flowering [DAYSBLOS] was $52 \pm 8$ days, and for the leaves, LEAFLENGTH was $10.1 \pm 2.0$ cm, and for the plants, PLANTHEIGHT was $326.0 \pm 138$ cm. For the pod traits of PODLENGTH, PODWIDTH, MATURITY, PODLOCULES and PODSEED, the means were $10.3 \pm 2.3$ cm, $17.4 \pm 2.0$ cm, $129 \pm 14$ days, $4 \pm 1$ and $3 \pm 1$, respectively. The means for the seed size traits were SEEDLENGTH of $20.6 \pm 2.0$ mm, SEEDHEIGHT of $8.8 \pm 1.0$ mm and SEEDWIDTH of $13.0 \pm 1.2$ mm. The minimum and maximum were 15.5 mm and 27.2 mm for SEEDLENGTH, 6.7 mm and 15.5 mm for SEEDHEIGHT, and 10.6 mm and 16.5 mm for SEEDWIDTH. For the 100-seed weight [SEEDHKM], the mean was $120.0 \pm 25.2$ g, while the minimum and maximum were 61.9 g and 203.3 g. The highest coefficient of variation was calculated for the trait PLANTHEIGHT (42.5%), and the lowest for SEEDWIDTH (9.0%).

**Table 1.** Summary statistics of the 12 quantitative morpho-agronomic traits of the runner beans (*Phaseolus coccineus* L.) in this collection from South-Eastern Europe.

| Trait Assessed | Code | Units | Range | Mean ± SD | CV (%) |
|---|---|---|---|---|---|
| Inflorescences | DAYSBLOS | days | 36–70 | 52 ± 8 | 15.1 |
| Leaves | LEAFLENGTH | cm | 4.0–14.0 | 10.1 ± 2.0 | 19.8 |
| Plants | HEIGHT | cm | 70.0–500.0 | 326.0 ± 138.0 | 42.5 |
| Pods | PODLENGTH | cm | 4.5–18.5 | 10.3 ± 2.3 | 22.4 |
| | PODWIDTH | mm | 11.9–22.9 | 17.4 ± 2.0 | 11.3 |
| | MATURITY | days | 101–170 | 129 ± 14 | 10.9 |
| | PODLOCULES | n | 2–7 | 4 ± 1 | 26.4 |
| | PODSEED | n | 1–6 | 3 ± 1 | 36.9 |
| Seeds | SEEDLENGTH | mm | 15.5–27.2 | 20.6 ± 2.0 | 9.6 |
| | SEEDHEIGTH | mm | 6.7–15.5 | 8.8 ± 1.0 | 11.5 |
| | SEEDWIDTH | mm | 10.6–16.5 | 13.0 ± 1.2 | 9.0 |
| | SEEDHKM | g | 61.9–203.3 | 120.0 ± 25.2 | 21.0 |

SD: standard deviation; CV: coefficient of variation.

All of the morpho-agronomic traits measured quantitatively or assessed qualitatively showed a wide range of variation among these runner-bean accessions from South-Eastern Europe. The frequency distributions for eight traits related to the inflorescence, leaf and plant data are shown in Figure 1. For the inflorescence trait of days to 50% flowering [DAYBLOS], 21 accessions were defined as early (≤45 days), 112 as medium (45–65 days) and six as late (≥66 days). The most abundant colours of the flower standards [BLOSVEX] among the 139 accessions were white (31 accessions) and carmine red (23). For the inflorescence trait of colour of the flower wings [BLOSWING], 37 accessions had white wings with lilac edges, and 30 had dark lilac wings with purple outer edges. The leaf traits included LEAFSHAPE, presence of anthocyanins in the leaf colour [LEAFANTHO], and terminal leaflet length [LEAFLENGTH]. The majority of these runner beans had a triangular leaf shape (100) and an absence of anthocyanins in the leaf colour (106). According to LEAFLENGTH, 25 accessions had short (≤8.4 cm), 76 medium (8.5–11.5 cm) and 38 long (≥11.6 cm) terminal leaflets. For the plant TYPE, almost all of these accessions (i.e., except three) showed indeterminate climbing plant growth. For the plant HEIGHT trait, 54 accessions grew to over 350 cm.

According to the 10 traits related to the pod data presented in Figure 2, 87 accessions had fully expanded immature pods [PODCOLIMM] with a normal green colour, and 25 had purple stripes with a green colour. For the pod suture strings [STRING], 94 accessions had moderately stringy pods, 36 had very stringy pods, and nine had just a few strings per pod. The most abundant pod colours at physiological maturity [PODCOLMAT] among the 139 accessions were persistent green (65 accessions) and yellow (44 accessions). According to PODLENGTH, 48 accessions were short (≤9.5 cm), 64 were medium (9.6–11.0 cm) and 26 were long (≥11.1 cm). According to PODWIDTH, 21 accessions were narrow (≤15.9 mm), 105 were medium (16.0–19.9 mm) and 13 were broad (≥20.0 mm). For the pod cross-section [PODCROSS], 88 accessions were pear-shaped and 47 were round elliptic. The pod curvature [PODCURV] was medium curved for 71 accessions and slightly curved for 62. For time to 90% pod maturity [MATURITY], 94 accessions were medium (90–130 days) and 45 accessions were late (≥131 days). According to the locules per pod [PODLOCULES], 63 accessions had three and 53 had four. For the seeds per pod [PODSEED], >85% of the accessions had two or three seeds per pod.

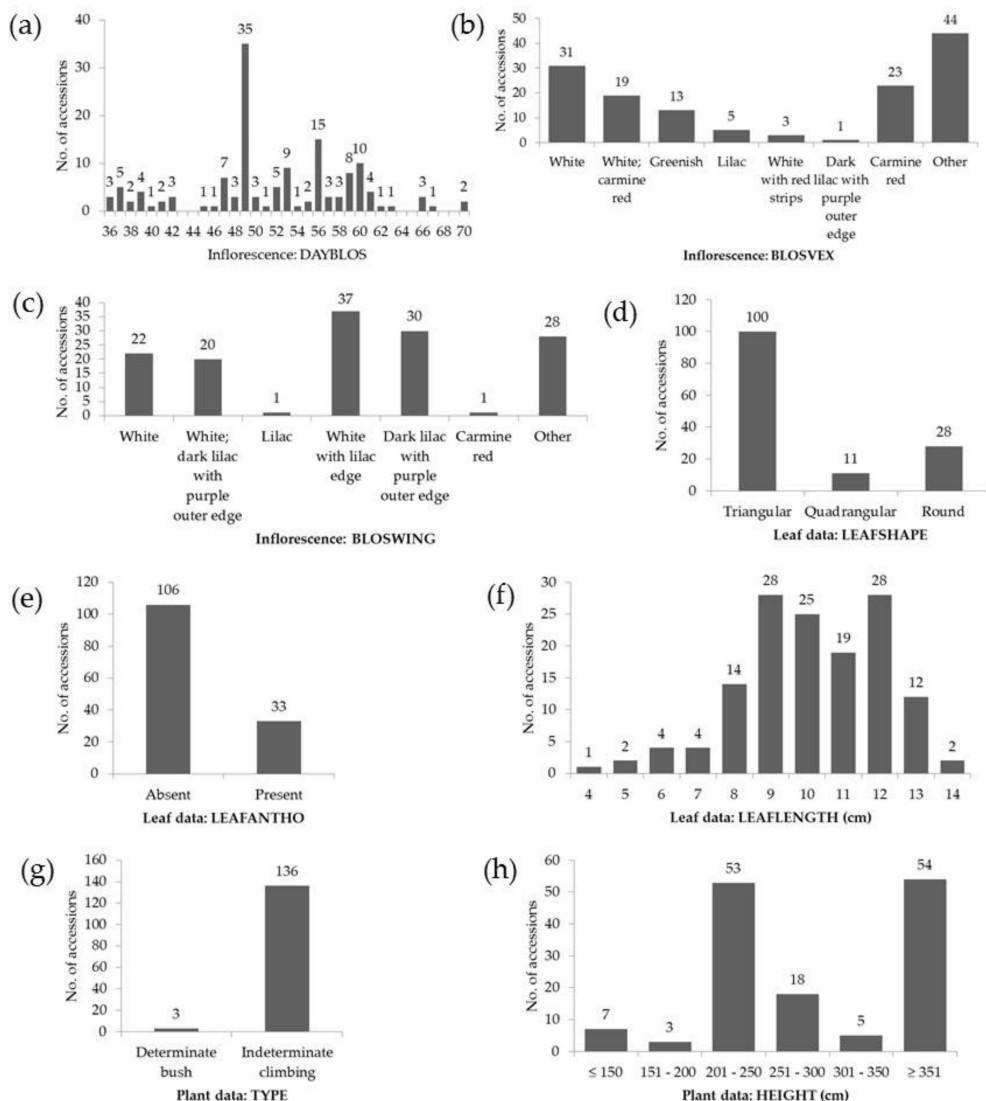

**Figure 1.** Frequency distributions for the runner-bean (*Phaseolus coccineus* L.) collection from South-Eastern Europe for eight traits related to inflorescences, leaves and plants. (**a**) Time from sowing to 50% flowering [DAYSBLOS]. (**b**) Colour of flower standard [BLOSVEX]. (**c**) Colour of flower wings [BLOSWING]. (**d**) Leaf shape [LEAFSHAPE]. (**e**) Leaf colour from anthocyanins [LEAFANTHO]. (**f**) Leaflet length [LEAFLENGTH]. (**g**) Plant growth habit [TYPE]. (**h**) Plant height [HEIGHT].

The frequency distributions of these runner beans for 10 traits related to their seed data are shown in Figure 3. According to the trait SEEDCOLOUR, 42 accessions had white seeds, 33 had cream and brown seeds (i.e., of two colours) and 28 had purple violet and black seeds (i.e., of two colours). The seed coat patterns [SEEDPATCHAR] were striped for 45 accessions, spotted for 43 and without a pattern for 43. For the brilliance of the seeds [BRILLIANCE], 86 accessions had shiny seeds and 37 had medium brilliant seeds. The seed shapes were oval for 61 accessions, cuboid for 49, kidney-shaped for 27 and markedly truncated for two. The colour of the seed hilum [SEEDHILUM] was white for 135 accessions and beige for four. According to the seed trait SEEDHILLUMRING for hilum ring colour, 49 accessions were white, 39 were brown, 27 were black and 23 were violet.

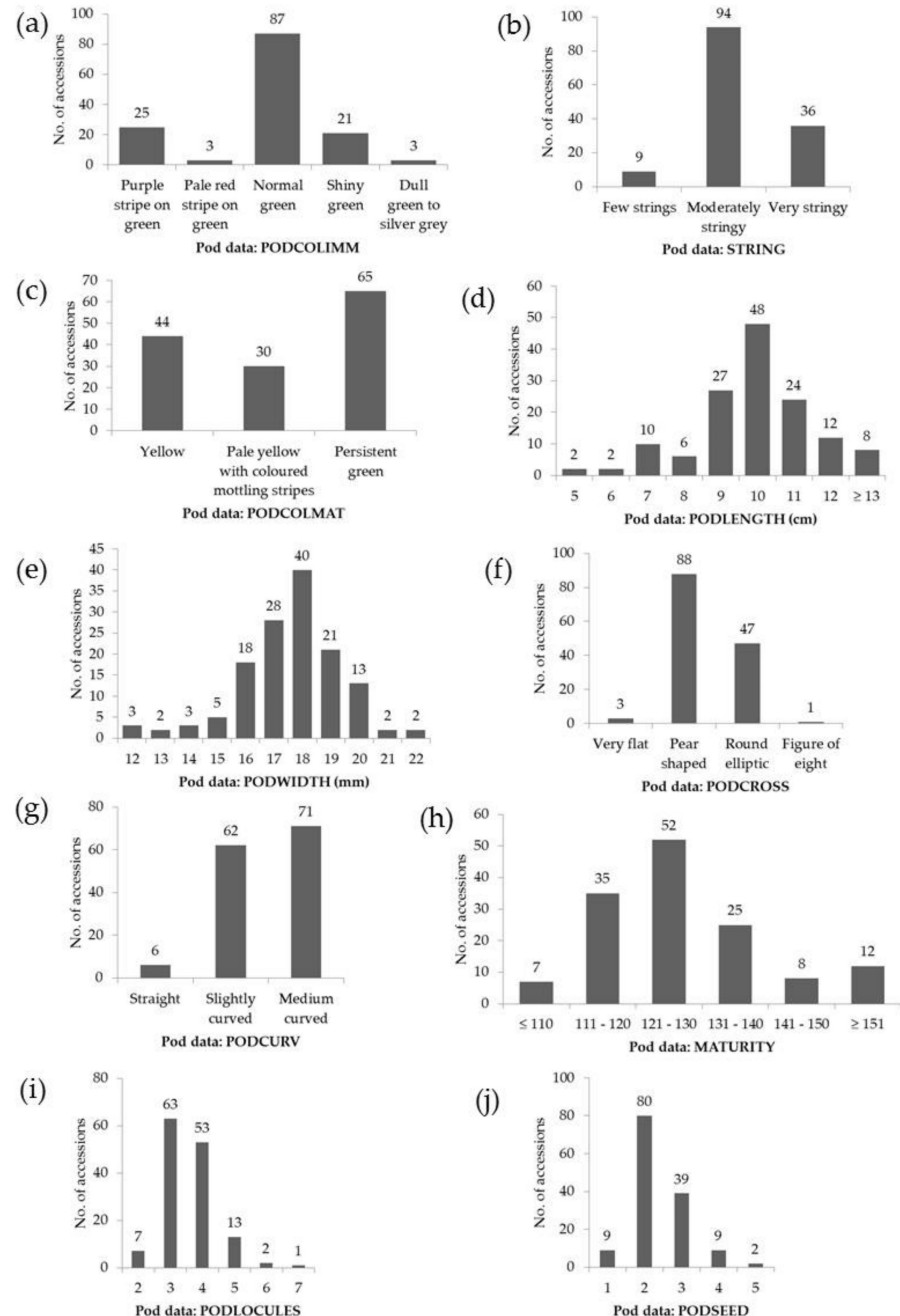

**Figure 2.** Frequency distributions for the runner-bean (*Phaseolus coccineus* L.) collection from South-Eastern Europe for 10 traits related to their pods. (**a**) Pod colour for fully expanded immature pods [PODCOLIMM]. (**b**) Pod suture strings [STRING]. (**c**) Pod colour at physiological maturity [PODCOLMAT]. (**d**) Pod length [PODLENGTH]. (**e**) Pod width [PODWIDTH]. (**f**) Pod cross-section [PODCROSS]. (**g**) Pod curvature [PODCURV]. (**h**) Pod time to 90% maturity [MATURITY]. (**i**) Number of locules per pod [PODLOCULES]. (**j**) Number of seeds per pod [PODSEED].

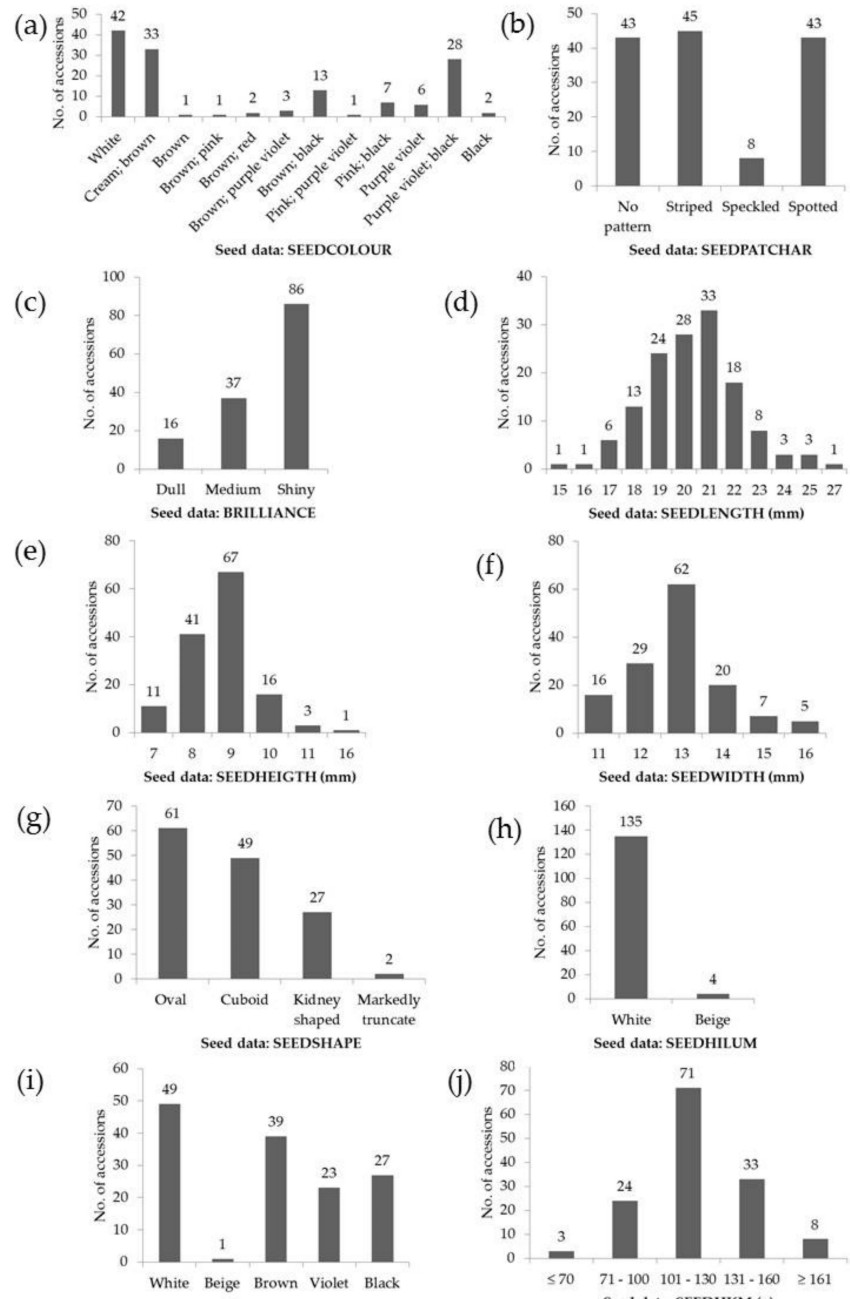

**Figure 3.** Frequency distributions for the runner-bean (*Phaseolus coccineus* L.) collection from South-Eastern Europe for 10 traits related to their seeds. (**a**) Seed coat colour [SEEDCOLOUR]. (**b**) Seed coat pattern [SEEDPATCHAR]. (**c**) Brilliance of seeds [BRILLIANCE]. (**d**) Seed length [SEEDLENGTH]. (**e**) Seed height [SEEDHEIGHT]. (**f**) Seed width [SEEDWIDTH]. (**g**) Longitudinal seed shape [SEEDSHAPE]. (**h**) Colour of seed hilum [SEEDHILUM]. (**i**) Colour of seed hilum ring [SEEDHILUMRING]. (**j**) One-hundred-seed weight [SEEDHKM].

Based on the quantitative seed measurements, these runner-bean accessions can be classified into three groups according to SEEDLENGTH, as small seeds of length ≤20.0 mm (58 accessions), medium seeds of length 20.1 mm to 24.9 mm (77) and long seeds of length ≥25.0 mm (4). According to SEEDWIDTH, 30 accessions had narrow seeds (≤12.0 mm), 101 had medium seeds (12.1–14.9 mm), and eight had broad seeds (≥15.0 mm). Similarly, the accessions can be classified into three groups according to the 100-seed weight [SEEDHKM], as low, medium and high (Figure 3). The low-weight

seeds had 100-seed weights ≤100.0 g (27 accessions), the medium-weight seeds were from 100.1 g to 179.9 g (109) and the high-weight seeds were ≥180.0 g (3).

The relationships between these 28 quantitative and qualitative morpho-agronomic traits for this runner-bean collection from South-Eastern Europe were evaluated using the Fisher's least significant difference method, to determine which traits or variables were significantly different. For the matrix developed for Pearson's correlation coefficients (Table 2), SEEDHILUMRING showed strong positive correlations with SEEDCOLOR1 (0.72, $p < 0.001$) and SEEDPATCHAR (0.71, $p < 0.001$), and moderate negative correlation with SEEDCOLOR2 (−0.64, $p < 0.001$). PODCURV had moderate positive correlations with HEIGHT (0.65, $p < 0.001$) and BLOSVEX1 (0.51, $p < 0.001$), and moderate negative correlation with PODCOLMAT (−0.53, $p < 0.001$). SEEDPATCHAR had moderate positive correlation with SEEDCOLOR1 (0.65, $p < 0.001$) and strong negative correlation with SEEDCOLOR2 (−0.78, $p < 0.001$). SEEDHKM had moderate positive correlations with SEEDLENGTH (0.62, $p < 0.001$) and SEEDWIDTH (0.60, $p < 0.001$). PODSEED had strong positive correlation with PODLOCULES (0.71, $p < 0.001$) and moderate positive correlation with PODLENGTH (0.52, $p < 0.001$). The other combinations of traits that showed strong positive correlations were between BLOSWING1 and BLOSVEX1 (0.75, $p < 0.001$), BLOSWING2 and BLOSVEX2 (0.85, $p <0.001$) and SEEDWIDTH and SEEDLENGTH (0.73, $p < 0.001$). Finally, the other combinations of traits that showed moderate positive or negative correlations were between LEAFANTHO and BLOSVEX1 (0.62, $p < 0.001$), HEIGHT and BLOSVEX1 (0.62, $p < 0.001$), PODCOLIMM and BLOSVEX1 (−0.60, $p < 0.001$), PODCOLMAT and HEIGHT (−0.69, $p < 0.001$), PODLOCULES and PODLENGTH (0.69, $p < 0.001$) and SEEDCOLOR1 and BLOSWING1 (0.51, $p < 0.001$).

Multiple regression analysis was performed for the 100-seed weight [SEEDHKM] versus the other traits, and this showed that SEEDWIDTH had the highest direct effects on SEEDHKM, followed by SEEDLENGTH, DAYSBLOS and SEEDHILUM, while for TYPE and MATURITY, SEEDHKM had a direct negative effect.

**Table 2.** Pearson correlation matrix between the morpho-agronomic traits of the runner-bean (*Phaseolus coccineus* L.) collection from South-Eastern Europe.

| | Trait | 1 | 2 | 3 | 4 | 5 | 6 | 7 | 8 | 9 | 10 | 11 | 12 | 13 | 14 | 15 | 16 | 17 | 18 | 19 | 20 | 21 | 22 | 23 | 24 | 25 | 26 | 27 | 28 | 29 | 30 |
|---|---|---|---|---|---|---|---|---|---|---|---|---|---|---|---|---|---|---|---|---|---|---|---|---|---|---|---|---|---|---|---|
| 1 | DAYSBLOS | | | | | | | | | | | | | | | | | | | | | | | | | | | | | | |
| 2 | BLOSVEX1 | 0.25[b] | | | | | | | | | | | | | | | | | | | | | | | | | | | | | |
| 3 | BLOSVEX2 | 0.17[a] | 0.26[b] | | | | | | | | | | | | | | | | | | | | | | | | | | | | |
| 4 | BLOSWING1 | 0.26[c] | 0.75[c] | 0.20[a] | | | | | | | | | | | | | | | | | | | | | | | | | | | |
| 5 | BLOSWING2 | 0.15 | 0.27[c] | 0.85[c] | 0.21[b] | | | | | | | | | | | | | | | | | | | | | | | | | | |
| 6 | LEAFSHAPE | −0.08 | −0.39[c] | −0.09 | −0.28[c] | −0.17[a] | | | | | | | | | | | | | | | | | | | | | | | | | |
| 7 | LEAFANTHO | 0.24[b] | 0.62[c] | 0.21[a] | 0.39[c] | 0.21[b] | −0.26[b] | | | | | | | | | | | | | | | | | | | | | | | | |
| 8 | LEAFLENGTH | 0.19[a] | −0.10 | −0.29[c] | −0.14 | −0.32[c] | 0.06 | −0.13 | | | | | | | | | | | | | | | | | | | | | | | |
| 9 | TYPE | 0.27[c] | 0.12 | −0.07 | 0.10 | −0.07 | 0.03 | 0.10 | 0.22[b] | | | | | | | | | | | | | | | | | | | | | | |
| 10 | HEIGHT | 0.48[c] | 0.62[c] | 0.31[c] | 0.47[c] | 0.32[c] | −0.40[c] | 0.39[c] | 0.00 | 0.19[a] | | | | | | | | | | | | | | | | | | | | | |
| 11 | PODCOLIMM | −0.12 | −0.60[c] | −0.11 | −0.31[c] | −0.11 | 0.22[b] | −0.33[c] | −0.02 | −0.14 | −0.38[c] | | | | | | | | | | | | | | | | | | | | |
| 12 | STRING | 0.07 | 0.37[c] | 0.15 | 0.20[a] | 0.16 | −0.14 | 0.28[c] | −0.03 | 0.01 | 0.35[c] | −0.08 | | | | | | | | | | | | | | | | | | | |
| 13 | PODCOLMAT | −0.18[a] | −0.43[c] | −0.32[c] | −0.39[c] | −0.33[c] | 0.34[c] | −0.27[b] | 0.21[b] | −0.01 | −0.69[c] | −0.10 | −0.42[c] | | | | | | | | | | | | | | | | | | |
| 14 | PODLENGTH | 0.23[b] | 0.11 | 0.09 | 0.17[a] | 0.09 | −0.01 | 0.03 | 0.19[a] | −0.06 | 0.21[b] | 0.07 | 0.18[a] | −0.25[b] | | | | | | | | | | | | | | | | | |
| 15 | PODWIDTH | 0.18[a] | 0.34[c] | 0.02 | 0.26[b] | 0.03 | −0.11 | 0.11 | 0.10 | 0.19[a] | 0.42[c] | −0.13 | 0.20[a] | −0.45[c] | 0.18[a] | | | | | | | | | | | | | | | | |
| 16 | PODCROSS | 0.38[c] | 0.31[c] | 0.08 | 0.35[c] | 0.12 | −0.19[a] | 0.29[c] | −0.02 | 0.25[b] | 0.30[c] | −0.30[c] | −0.01 | −0.07 | 0.14 | 0.05 | | | | | | | | | | | | | | | |
| 17 | PODCURV | 0.35[c] | 0.51[c] | 0.25[b] | 0.37[c] | 0.26[b] | −0.35[c] | 0.33[c] | −0.07 | 0.32[c] | 0.65[c] | −0.31[c] | 0.31[c] | −0.53[c] | 0.09 | 0.32[c] | 0.34[c] | | | | | | | | | | | | | | |
| 18 | MATURITY | 0.34[c] | −0.25[b] | 0.13 | −0.14 | 0.13 | 0.26[b] | −0.09 | −0.22[b] | −0.06 | −0.17[a] | 0.26[b] | −0.09 | 0.18[a] | 0.25 | −0.21[a] | 0.02 | −0.22[b] | | | | | | | | | | | | | |
| 19 | PODLOCULES | 0.08 | −0.15 | −0.02 | −0.05 | −0.02 | 0.13 | −0.12 | 0.21[b] | −0.17[a] | −0.14 | 0.21[b] | 0.07 | 0.04 | 0.69[c] | −0.08 | 0.04 | −0.23[b] | 0.08 | | | | | | | | | | | | |
| 20 | PODSEED | −0.02 | −0.19[a] | 0.05 | −0.11 | 0.04 | 0.20[a] | −0.16 | −0.05 | −0.22[b] | −0.23[b] | 0.18[a] | 0.01 | 0.07 | 0.52[c] | −0.09 | −0.01 | −0.22[b] | 0.25[b] | 0.71[c] | | | | | | | | | | | |
| 21 | SEEDCOLOR1 | 0.00 | 0.33[c] | −0.05 | 0.51[c] | −0.03 | 0.03 | 0.25[b] | −0.11 | 0.09 | −0.02 | −0.17[a] | 0.09 | 0.04 | −0.07 | 0.02 | 0.08 | 0.04 | −0.09 | −0.13 | −0.14 | | | | | | | | | | |
| 22 | SEEDCOLOR2 | −0.12 | −0.45[c] | 0.13 | −0.31[c] | 0.10 | −0.02 | −0.26[b] | −0.22[b] | −0.24[b] | −0.13 | 0.34[c] | −0.11 | −0.02 | 0.01 | −0.13 | −0.16[a] | −0.11 | 0.20[a] | 0.16[a] | 0.14 | −0.47[c] | | | | | | | | | |
| 23 | SEEDPATCHAR | −0.08 | 0.33[c] | −0.07 | 0.36[c] | −0.04 | 0.07 | 0.20[a] | −0.03 | 0.15 | −0.08 | −0.09 | 0.08 | 0.04 | −0.13 | 0.11 | 0.02 | 0.02 | −0.16[a] | −0.20[a] | −0.14 | 0.65[c] | −0.78[c] | | | | | | | | |
| 24 | BRILLIANCE | −0.19[a] | 0.07 | −0.21[a] | 0.12 | −0.22[b] | 0.04 | −0.08 | 0.24[b] | −0.08 | −0.05 | 0.06 | 0.14 | −0.07 | 0.13 | 0.10 | −0.14 | −0.10 | −0.42[c] | 0.15 | 0.02 | 0.30[c] | −0.32[c] | 0.40[c] | | | | | | | |
| 25 | SEEDLENGTH | 0.08 | 0.03 | −0.04 | 0.09 | −0.06 | 0.12 | 0.09 | 0.09 | 0.06 | 0.01 | 0.00 | 0.07 | 0.05 | 0.29[c] | 0.13 | −0.02 | 0.01 | 0.09 | −0.12 | −0.01 | 0.18[a] | −0.19[a] | 0.16[a] | −0.01 | | | | | | |
| 26 | SEEDHEIGHT | 0.15 | 0.05 | −0.09 | −0.01 | −0.09 | 0.01 | 0.29 | 0.27 | 0.45 | 0.87 | −0.15 | 0.05 | 0.07 | 0.15 | 0.09 | 0.12 | 0.12 | 0.03 | −0.01 | −0.06 | 0.04 | −0.21[b] | 0.08 | 0.07 | 0.36[c] | | | | | |
| 27 | SEEDWIDTH | 0.25[b] | 0.19[a] | 0.02 | 0.17[a] | 0.00 | 0.00 | 0.11 | 0.10 | 0.15 | 0.28[c] | −0.08 | 0.11 | −0.14 | 0.19[a] | 0.39[c] | −0.02 | 0.17[a] | 0.09 | −0.19[a] | −0.11 | 0.09 | −0.19[a] | 0.14 | −0.06 | 0.73[c] | 0.42[c] | | | | |
| 28 | SEEDSHAPE | 0.08 | −0.17[a] | 0.07 | −0.05 | 0.09 | 0.09 | −0.02 | 0.00 | −0.10 | −0.10 | 0.34c | 0.02 | −0.07 | 0.32[c] | −0.07 | 0.32[c] | −0.07 | −0.15 | 0.23[b] | 0.25[b] | 0.34[c] | −0.04 | 0.17[a] | −0.07 | 0.01 | 0.22[b] | −0.09 | 0.17[a] | | |
| 29 | SEEDHILUM | −0.11 | −0.11 | −0.19[a] | −0.09 | −0.18[a] | 0.09 | −0.09 | 0.05 | −0.20[a] | −0.14 | 0.00 | −0.07 | 0.16 | −0.14 | −0.18[a] | −0.18[a] | −0.22[b] | −0.02 | −0.09 | −0.12 | 0.13 | −0.11 | 0.06 | −0.04 | −0.01 | −0.10 | −0.10 | −0.08 | | |
| 30 | SEEDHILUMRING | −0.05 | 0.36[c] | −0.11 | 0.42[c] | −0.05 | 0.00 | 0.18[a] | 0.06 | 0.12 | 0.04 | −0.26[b] | 0.08 | −0.01 | 0.00 | 0.20[a] | 0.08 | 0.14 | −0.30[c] | −0.13 | −0.14 | 0.72[c] | −0.64[c] | 0.71[c] | 0.37[c] | 0.14 | 0.06 | 0.08 | −0.16 | −0.14 | |
| 31 | SEEDHKM | 0.14 | 0.09 | −0.03 | 0.10 | −0.05 | −0.05 | 0.03 | 0.17[a] | −0.09 | 0.14 | −0.06 | 0.15 | −0.11 | 0.31[c] | 0.29[c] | −0.03 | 0.15 | −0.15 | 0.00 | −0.03 | 0.07 | −0.16 | 0.08 | 0.11 | 0.60[c] | 0.44[b] | 0.62[c] | 0.04 | 0.07 | 0.15 |

[a], $p < 0.05$; [b], $p < 0.01$; [c], $p < 0.001$. DAYSBLOS, days from sowing to 50% flowering; BLOSVEX1(2), colour of flower standard; BLOSWING1(2), colour of flower wings; LEAFSHAPE, leaf shape; LEAFANTHO, leaf colour from anthocyanins; LEAFLENGTH, leaflet length; TYPE, plant growth habit; HEIGHT, plant height; PODCOLIMM, pod colour from fully expanded immature pod; STRING, pod suture strings; PODCOLMAT, pod colour at physiological maturity; PODLENGTH, pod length; PODWIDTH, pod width; PODCROSS, pod cross-section; PODCURV, pod curvature; MATURITY, days to 90% maturity; PODLOCULES, locules per pod; PODSEED, seeds per pod; SEEDCOLOUR1(2) seed coat colour; SEEDPATCHAR, seed coat pattern; BRILLIANCE, brilliance of seed; SEEDLENGTH, seed length; SEEDHEIGHT, seed height; SEEDWIDTH, seed width; SEEDSHAPE, longitudinal seed shape; SEEDHILUM, colour of seed hilum; SEEDHILUMRING, colour of seed hilum ring; SEEDHKM, 100 seed weight.

### 3.2. Comparison of Runner-Bean Variability among the South-Eastern European Countries

The variability of these 28 morpho-agronomic traits for this runner-bean collection across the five South-Eastern European countries of Slovenia, Bosnia and Herzegovina, Serbia, North Macedonia and Romania were determined. The highest variability was seen between the countries (76.39%, $p < 0.001$; *F*-ratio, 508.21), which are representative of the different geographic origins, while the variability within the countries was 23.61% ($p < 0.001$). In particular, for the within-country variability (Table 3), the Slovenian and Serbian accessions showed the highest (44.96%, 26.28%, respectively). Lower within-country variabilities were seen for the accessions from North Macedonia (9.64%), Romania (9.10%) and Bosnia and Herzegovina (3.01%). Interestingly, although most of the accessions in the collection originated from North Macedonia (64 accessions; 46%), the variability between these was relatively low.

**Table 3.** Variability of the runner-bean (*Phaseolus coccineus* L.) collection within the five originating South-Eastern European countries.

| Country of Origin | Number of Accessions | Variability (%) | *F*-Ratio |
|---|---|---|---|
| Slovenia | 44 | 44.96 *** | 58.19 |
| Bosnia and Herzegovina | 10 | 3.01 *** | 366.65 |
| Serbia | 9 | 26.28 *** | 31.89 |
| North Macedonia | 64 | 9.64 *** | 639.44 |
| Romania | 12 | 9.10 *** | 144.43 |

***, $p < 0.001$.

Based on the principal component analysis performed on the 28 morpho-agronomic traits for this runner-bean collection, a three-dimensional score plot was created, as presented in Figure 4. The principal component analysis was performed for a data structure study over a reduced dimension, to provide the maximum amount of information from the data. The first eight principal components accounted for 68.93% of the total variation for these 28 morpho-agronomic traits. The relative contribution to this total variance of Component 1 was 18.97%, with the major contributory traits (in descending order) of BLOSVEX1, HEIGHT, BLOSWING1, PODCURV and LEAFANTHO. The relative contribution of Component 2 was 11.85% of the total variance, mainly according to SEEDPATCHAR, SEEDHILUMRING, SEEDCOLOUR2 and SEEDCOLOUR1. Finally, the relative contribution of Component 3 was 9.85% of the total variance, mainly according to PODLENGTH, SEEDLENGTH, DEEDHKM and SEEDWIDTH.

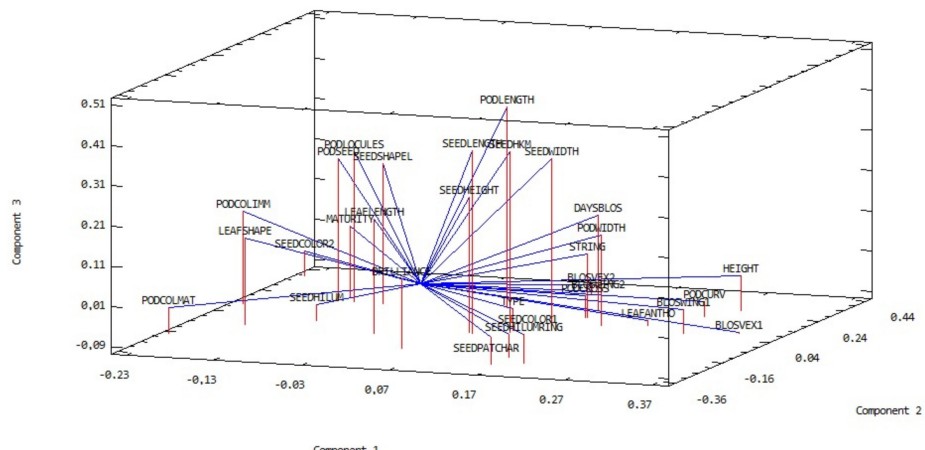

**Figure 4.** Three-dimensional score plot from the principal component analysis for the relative distributions of the 28 morpho-agronomic traits for this runner-bean (*Phaseolus coccineus* L.) collection from South-Eastern Europe. For trait codes, see Table 2.

The results of the cluster analysis (Ward's method, squared Euclidean) according to the 28 morpho-agronomic traits are shown in Figure 5. This divided the collection into three main groups. Group 1 (50 accessions) included all 12 of the accessions from Romania, 20 from North Macedonia, six from Slovenia, and four from Serbia. Additionally, Group 1 included the commercial varieties 'Bonela' from the trials in Slovenia, Serbia, and Bosnia and Herzegovina, and 'Darko' from the trials in Slovenia, Serbia and North Macedonia. Group 2 included most of the accessions that originated from Slovenia (38), all of those from Bosnia and Herzegovina (10), five accessions from Serbia, and one from North Macedonia. Additionally, Group 2 included the commercial varieties 'Emergo' from all of the five country trials, 'Bonela' from the trials in Romania and North Macedonia, and 'Darko' from the trials in Romania. Group 3 comprised only the accessions that originated from North Macedonia (43).

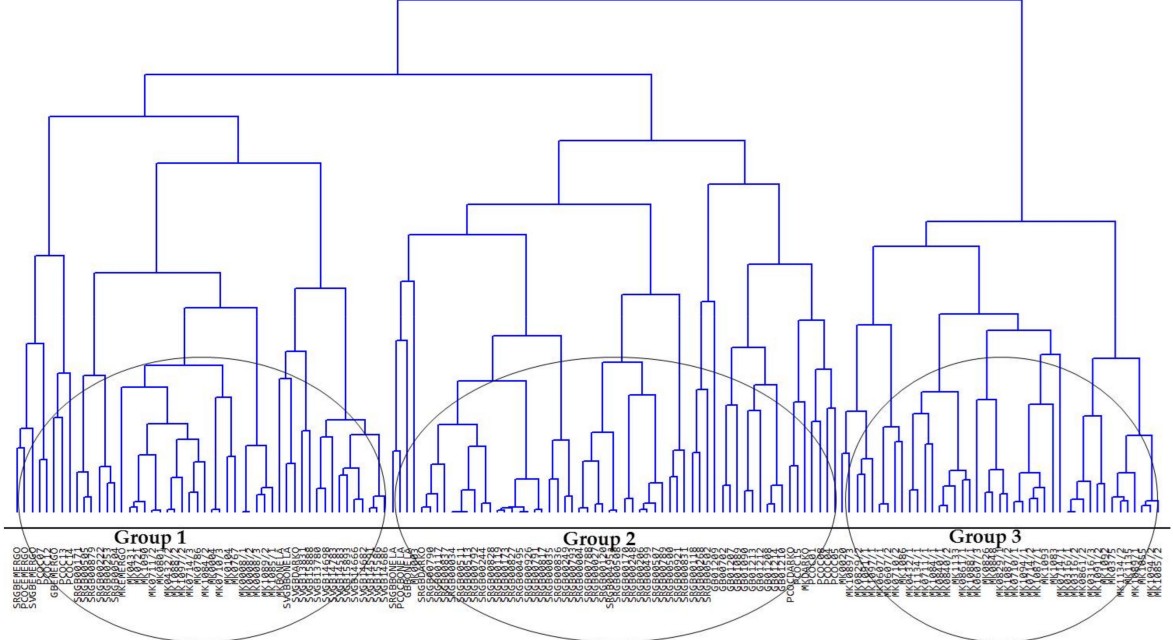

**Figure 5.** Cluster analysis (Ward's method, squared Euclidean) according to the 28 morpho-agronomic traits of this runner-bean (*Phaseolus coccineus* L.) collection from South-Eastern Europe. For individual accession codes, see Supplementary Table S1.

## 4. Discussion and Conclusions

Food legumes, including the runner bean, are an important part of the human diet; moreover, they support environmental sustainability through biological symbiotic fixation of nitrogen [26,27]. In South-Eastern Europe, the runner bean represents a nutritional and environmental resource that is well adapted to the specific growing conditions and microclimate agro-environments. This germplasm should be genetically preserved and improved for efficient future use. Indeed, evaluation of runner-bean accessions might reveal potentially valuable traits that are rare or cannot be found in common bean germplasm today, and would thus be useful for genetic improvement of the common bean.

The present study started with multiplication and conservation of the runner-bean germplasm from five countries in South-Eastern Europe (with each representative of a specific geographic origin): Slovenia, Bosnia and Herzegovina, Serbia, North Macedonia and Romania. Furthermore, we then carried out one of the first large-scale morpho-agronomic characterisations of the runner-bean collections conserved in South-Eastern Europe, using the approach exemplified by the studies of Rodino et al. [28] and Zeven et al. [29]. This germplasm represents a wide range of genetic variability for the different morpho-agronomic traits studied for these 142 runner-bean accessions.

The frequency distribution obtained for 28 morpho-agronomic traits allowed differentiation and classification of all of these accessions. Based on the inflorescence trait of days to 50% flowering,

the majority of these accessions (>80%) fell into the medium group of flowering, while for the pod trait of days to 90% pod maturity, 68% of the accessions fell into the medium maturity period and 32% into the late maturity period. There are three botanical varieties of runner bean, as the white-flowered type of var. *albiflorus*, the red-flowered type of var. *coccineus* and the type with white and red flowers of var. *bicolor* [8]. Here, the most abundant flower colours were white (22%), carmine red (16%) and white and red flowers (14%). As reported by Rodriguez et al. [8], the European geographic distribution of 331 runner-bean varieties they studied showed that var. *bicolor* accessions were prevalent in Central Europe, var. *albiflorus* in Southern Europe and var. *coccineus* in Northern Europe.

Among the different traits that have been defined for common bean, the seed traits are the most important as they provide the major determinants of the commercial acceptability of the varieties [30–33]. Distribution classes for seed length and 100-seed weight had been chosen based on previous reports [1,14,34]. Based on seed length, the present runner-bean accessions were classified into three groups, as short (≤20.0 mm; 42%), medium (20.1 to 24.9 mm; 55%) and long seeds (≥25.0 mm; 3%). According to data for seed width here, 22% were narrow (≤12.0 mm), 73% medium (12.1–14.9 mm) and 5% broad (≥15.0 mm). Then for 100-seed weight, these runner bean accessions were classified into the three groups of low (≤100.0 g; 20%), medium (100.1–179.9 g; 78%) and high (≥180.0 g; 2%). These data are in agreement with previous reports for runner-bean accessions from Slovenia [14] and Romania [34]. For the trait of seed colour, the majority of these present runner-bean accessions belonged to the mixed colour class (i.e., of two colours). The most abundant seed colours were white (30%), cream and brown (24%) and purple violet and black (20%). Over one third of these runner-bean accessions had a striped coat pattern, followed by a spotted coat, and by seeds without any pattern. The majority of seeds were oval shaped, followed by cuboid and kidney shaped.

Correlations between all of the morpho-agronomic traits for this runner-bean collection from South-Eastern Europe were also investigated. These data indicated strong positive correlations for the colours of the flower wings versus standard, and of the seed hilum ring versus both coat colour and pattern, for seed width versus length and seeds versus locules per pod. Instead, there was a strong negative correlation for seed coat pattern versus colour. Overall, the highest correlations were seen within the fluorescence, seed and pod traits. Multiple regression analysis for 100-seed weight versus the other traits showed that seed width had the highest positive direct effects on seed weight, followed by seed length, days to 50% flowering and colour of seed hilum; in contrast, plant growth habit and days to 90% pod maturity had negative direct effects on seed weight.

Although the variability of common-bean accessions from central and South-Eastern Europe has already been reported in terms of morphological and molecular markers [35–38], such information on the runner bean from this region was lacking. Recently, runner-bean seeds of a Slovenian collection were characterised morphologically [14], although there remain few data for other traits, and none for runner beans from the other South-Eastern European countries covered here. However, a limited number of phenotypic traits were evaluated in a genetic diversity study as part of an investigation into a large set of accessions from all over Europe (including Slovenia and Romania) using chloroplast microsatellites [8].

The variability of the present runner-bean collection was assessed using 28 morpho-agronomic characteristics. The highest variability was seen between the countries, as representative of the different geographic origins of these accessions, with lower variability within the countries. The Slovenian and Serbian runner-bean accessions showed the highest variabilities here, with considerably lower variability seen for the accessions from North Macedonia, even though the large majority of these runner-bean accessions were from North Macedonia. Narrower genetic diversity was also previously seen for common-bean accessions from North Macedonia when compared to accessions from four other countries, on the basis of molecular markers, i.e., for Serbia, Bosnia and Herzegovina, Croatia and Slovenia [37]. In this previous study on common bean, similarly high levels of genetic diversity (as estimated by the number of alleles, number of effective alleles, Shannon's normalisation index and expected heterozygosity) were seen for the accessions from Slovenia and Serbia, and also from Bosnia

and Herzegovina. In contrast, in the present study, the lowest variability for these runner beans was seen for the accessions from Bosnia and Herzegovina.

Cluster analysis based on these collected morpho-agronomic data classified the present runner-bean accessions into three groups, which might indicate their gene pools of origin, i.e., the Mesoamerican gene pool, the Andean gene pool and a group with putative hybrids between these two gene pools. The grouping of common-bean accessions in terms of their countries of origin was reported by Maras et al. [37], which might be similarly indicative of the origins for these runner-bean accessions. Based on the results of the cluster analysis here, most of the Slovenian and Serbian runner-bean accessions, and all of those from Bosnia and Herzegovina were grouped into the same cluster: Group 2. On the other hand, the majority of the Macedonian accessions (43 of 64 accessions) formed a separate cluster, Group 3, where no accessions from the other countries were included. This suggests that the selection in the past was oriented towards similar types of varieties particularly in Macedonia, i.e., the varieties with white seeds, which are traditionally used in Macedonian cuisine.

Morpho-agronomic information in this study will help to promote runner-bean breeding for the development of new varieties with favourable traits. To date, only limited information is available on runner-bean accessions from South-Eastern Europe [14,37,39], as compared to the common bean [14,35–38,40]. As a further step, analysis of this diversity using molecular approaches is recommended, to provide more information on the overall genetic diversity, and potentially on the genes responsible for the specific morpho-agronomic traits in this runner-bean collection from South-Eastern Europe.

**Supplementary Materials:** The following are available online at http://www.mdpi.com/2071-1050/11/21/6165/s1. **Table S1**. The runner-bean (*Phaseolus coccineus* L.) accessions included in this morpho-agronomic characterisation from the 2017 growing season. **Table S2**. Morpho-agronomic traits included in the study [25]. **Table S3**. Summary data for morpho-agronomic traits according to the countries of origin of the three commercial varieties included in the study. **Table S4**. Summary statistics for the 12 quantitative characteristics in the 139 accessions (and three varieties) of runner bean (*Phaseolus coccineus* L.) as the individual country evaluations. **Figure S1**. Box and whisker plot for the 28 morpho-agronomic traits of this runner-bean (*Phaseolus coccineus* L.) collection from South-Eastern Europe. **Figure S2**. Climate conditions at the five experimental sides during the growing season in 2017 as obtained from the national meteorological data services.

**Author Contributions:** Conceptualization, B.P. and V.M.; Data curation, M.V., M.A., S.I., C.B. and J.Š.-V.; Formal analysis, L.S. and B.P.; Funding acquisition, C.B. and V.M.; Investigation, M.V., M.A., S.I. and C.B.; Methodology, B.P. and J.Š.-V.; Resources, M.V., M.A., S.I., C.B. and J.Š.-V.; Software, L.S. and B.P.; Supervision, V.M.; Visualization, L.S.; Writing—original draft, L.S.; Writing—review and editing, V.T., S.I. and V.M.

**Funding:** This study was funded by grants from the Slovenian Research Agency, Ljubljana, Slovenia, titled "Agrobiodiversity" (P4-0072), and from the European Cooperative Programme for Plant Genetic Resources (ECPGR) titled, "Efficient Management of Genetic Resources for Smart Legumes Utilisation".

**Conflicts of Interest:** The authors declare that they have no conflicts of interest.

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
