# Peer review of "Morpho-Agronomic Characterisation of Runner Bean (Phaseolus coccineus L.) from South-Eastern Europe"

_sustainability, doi:10.3390/su11216165_

Round 1

Reviewer 1 Report

The present study investigated the morpho-agronomic traits of runner-bean (Phaseolus coccineus L.) from south-eastern Europe.
The results are somewhat interesting and this work may be publishable in this journal, and results may be an useful genetic information for plant breeders for the breeding of new bean varieties.

Reviewer 2 Report

Dear authors, I would like you to reconsider your article, which in my opinion suffers from some over-simplification in the scientific approach and in the management of experimental data.

It is certainly interesting to have collected a huge collection of local accessions / populations of 5 different nations (139 + 3 accessions, plus than 17.000 plants is not a small phenotyping effort), but the work seems to omit the phenotypic discordance that cannot but exist when the same materials are characterized in such different and distant environments.

Why is this discrepancy not reported? Reading your text it seems that in all 5 experimental field you have collected the same qualitative and quantitative values, without the slightest variation among the sites. It seems really difficult for me to imagine that 5 characterizations conducted in different areas, with different climates, with maybe different irrigation supplies, with unknown seeding dates, etc etc could have caused such a high number concordance (same DAYSBLOS or HEIGHT, SEEDHKM or MATURITY??? just to name a few). Not to mention the fact that presumably the data was collected by different operators and it is easy for different operators to have different sensitivities (for example in the collection of qualitative data such as flowers or pods colors).

As regards the use of local populations with a presumable / potential degree of genetic variability within the population itself, no preliminary work is cited to reduce the same variability (for example a selection made in previous years) in order to collect more reliable data but no internal variability among 24 x 5 = 120 measured plants is mentioned. In this way the environmental effects are not estimated, let alone the genetic ones or the hereditability, and so no interesting accessions are described for the plant breeders for the breeding of new runner bean varities.

As regards the experimental data only the average of the average values ​​with an average standard deviation is reported, not even in the supplementary data.

Also the description of the value distributions is made in an arbitrary manner (for example, why does SEEDLENGHT have as extremes 20.0 mm and 25.0 mm?) On the basis of which works / articles have the distribution classes been chosen which then justify the discussion of the different parameters/descriptors?

Another consideration that is missing, apart from the association of macro chromatic characters with cluster groups 1, 2 and 3, is whether some parameters are more or less present within the same clusters (for example, larger sizes of leaves in the group 1 or quadrangular form in cluster group 2?).

Furthermore, I suggest a descriptive table instead of the text (lines 130-159) where the descriptors are outlined.

Finally, it seems to me that the discussion of the results may be a little more "discussed".

Reviewer 3 Report

This is a very well written manuscript. The manuscript fits within the scope of the journal. The manuscript is interesting and the idea is very nice.  The title is clear and it is adequate to the content of the article. The study methods and results are explained clearly. The conclusions or summary are supported by the content. The author’s work on discussing achieved results is appreciated.

I have one recommendation for authors: Please highlight the degree of novelty and originality of the work.

Round 2

Reviewer 2 Report

I have read the corrections to the text and the reply letter from the authors, including the addition of a table to supplementary materials which further clarifies the paucity of the work submitted.

First of all, they have characterized 5 collections of germplasm, but each of these only in their country of origin, without repeating the work in the other 4 different states! For me this is unacceptable, the estimate of the environmental factor is completely lacking and, therefore, it makes no sense to put together all the bio-morphological parameters and derive a study from them.

In fact, after my request for clarification, because in the first draft it was not clear that they had cultivated, for example, the 44 Slovenian accessions only in Slovenia,10 accessions from Bosnia only in Bosnia, etc etc, they added a characterization table of 3 varieties that they phenotyped in the 5 different sites. I attached this table in this email and if you give a look to this doc, you can see highlighted as most of the parameters present anomalies, errors of assessment or quantitative or qualitative differences due to the different environments. And if there are many differences for 3 cultivars found between the 5 phenotyping sites, who knows how many errors, anomalies, quantitative or qualitative differences would be to be found if all accessions were characterized together. This discrepancy is completely omitted, as if it were not important, but in this way you can not properly distinguish amongst the accessions. For example the cultivar Emergo was 70 +- 20 cm height in Bosnia and 500+-80 cm in Macedonia (TYPE HEIGHT), or “Bonela” weight of 100 seeds was 193.1 g in Slovenia and only 106.8 in Romania… DAYSBLOS are always different, as well are MATURITY and or the most of the traits regarding pods or seeds, etc.

In the preliminary evaluation I had asked the authors explanation if they had measured the environmental component, still they had not clarified the trick of the "national" cultivation, but they answered: “Phenotyping discordance of course exists, but it is necessary to indicate that all of the evaluators within the consortium countries are well trained and experienced for evaluation plant materials from gene banks…” and this mantra has been repeated several times.

Reviewer 3 Report

The authors have addressed all of my concerns.
